# OpenReview forum: "Binary Hypothesis Testing for Softmax Models and Leverage Score Models"
_ICML.cc/2025/Conference — ICML 2025 poster_

### Official Review · Reviewer_acsd · 2025-03-09

**Overall Recommendation:** 2

**Summary:**

The paper addresses the problem of binary hypothesis testing in the context of two important probabilistic models: softmax models and leverage score models. The main contributions and findings of the paper are as follows:

- **Binary Hypothesis Testing for Softmax Models**: The authors study the fundamental problem of determining which one of two given softmax models is the true model, based on queries to the models. They establish that the sample complexity for this task is asymptotically $O(ϵ^{−2})$, where ϵ quantifies a specific distance between the parameters of the two models.

- **Connection to Leverage Score Models**: The paper draws an analogy between softmax models and leverage score models, which are widely used in algorithmic applications such as linear algebra and graph theory.

- **Binary Hypothesis Testing for Leverage Score Models**: The authors extend their analysis to leverage score models and derive similar results for binary hypothesis testing in this setting.

**Claims And Evidence:**

The claims in the paper are generally supported by clear and convincing evidence, with rigorous mathematical derivations and proofs provided for the main results. The sample complexity bounds for both softmax and leverage score models are substantiated through formal theorems, such as Theorem 3.1 (general result for softmax models) and Theorem 4.1 (general result for leverage score models). Lower bounds (e.g., Theorem 3.2, Theorem 4.2) and upper bounds (e.g., Theorem 3.5, Theorem 4.3). Additionally, the analogy between softmax and leverage score models is well-motivated by their shared structural properties, and the energy constraints on inputs are justified to avoid trivial solutions.

**Essential References Not Discussed:**

None

**Experimental Designs Or Analyses:**

N/A

**Methods And Evaluation Criteria:**

The methods proposed in the paper are well-suited to the problem of binary hypothesis testing for softmax and leverage score models. The authors use mathematically rigorous frameworks, such as Hellinger distance, to define sample complexity bounds, ensuring theoretical soundness. The evaluation focuses on deriving tight lower and upper bounds for sample complexity, which are validated through formal proofs, demonstrating their relevance to distinguishing between parameterized models.

**Other Comments Or Suggestions:**

- **Broader Context**: The discussion of related work is thorough but could better emphasize how this paper’s contributions differ from prior studies on hypothesis testing for machine learning models.

- **Typographical Errors**:

  - In Section 1, “Then a question arose:” is stylistically abrupt and could be rephrased for a smoother transition.
  - Ensure proper formatting of references (e.g., missing publication in citations like “Brown et al., 2020;?”).

**Other Strengths And Weaknesses:**

**Strengths:**

The paper demonstrates several strengths in terms of originality, significance, and clarity. The paper offers a novel theoretical framework for binary hypothesis testing in softmax and leverage score models. The authors provide detailed theoretical analysis, including tight bounds on sample complexity and rigorous proofs, which contribute to the significance of the work. Additionally, the analogy drawn between softmax and leverage score models bridges concepts from machine learning and linear algebra. The paper’s rigorous mathematical formulations highlights its originality and relevance.

**Weakness**:

The paper has several weaknesses that limit its overall impact and coherence. The primary issue lies in the disconnect between the stated motivation—understanding large language models (LLMs) through the softmax attention mechanism—and the actual focus of the work, which is on sample complexity for binary hypothesis testing of softmax distributions. The analysis appears tangential to the original motivation, and the results are not tied back to improving theoretical or practical understanding of LLMs. The paper primarily focuses on asymptotic results without empirical validation, which limits its practical applicability. Additionally, the paper does not deeply explore connections to related work or practical advancements in LLMs. Similarly, the conclusions and future work primarily address hypothesis testing problems without exploring meaningful real-world applications or implications for LLMs. Finally, the paper’s structure could be improved.

**Questions For Authors:**

The questions for the authors are already addressed in the "Weaknesses" section.

**Relation To Broader Scientific Literature:**

The paper's key contributions are closely tied to broader scientific literature in machine learning, linear algebra, and statistical hypothesis testing. The analogy drawn between softmax models and leverage score models connects the work to established research in numerical linear algebra and graph theory, where leverage scores are widely used for tasks like graph sparsification, maximum matching, and optimization problems. The binary hypothesis testing framework leverages classical results in hypothesis testing extending these ideas to structured models like softmax and leverage scores.

**Theoretical Claims:**

The paper provides rigorous proofs for its theoretical claims, particularly regarding the sample complexity of binary hypothesis testing for softmax and leverage score models. The main results, such as the asymptotic sample complexity bounds of $O(ϵ^{−2})$ and $Ω(ϵ^{−2})$, are supported by formal derivations using tools like Hellinger distance and variance-based metrics (e.g., Theorems 3.1, 3.2, 3.5 for softmax models and Theorems 4.1, 4.2, 4.3 for leverage score models).

---

> ### Author Rebuttal · Authors · 2025-04-01
>
> We express our deepest gratitude to the reviewer for the time and effort in reviewing our work. Below, we want to respond to the weaknesses and questions.
>
> **Concern 1**: The primary issue lies in the disconnect between the stated motivation—understanding large language models (LLMs) through the softmax attention mechanism—and the actual focus of the work, which is on sample complexity for binary hypothesis testing of softmax distributions.
>
> **Answer**: We thank you very much for your insightful comments. In Sections 3 and 4, we study the distinguishability of models (softmax and leverage score based) through the lens of binary hypothesis testing, establishing tight sample complexity bounds. These results directly address the challenge of determining how much information (or how many queries) is needed to tell apart closely related models, a theoretical formulation aligned with understanding model "abilities" via limited parameter access. Moreover, our framework sets a path toward identifying distinguishable components of large models. For instance, showing that certain parameters contribute more significantly to distinguishability (via Hellinger distance bounds) offers insight into what might constitute an “ability region” within a model, aligning with the introductory motivation.
>
> Thank you very much for your help checking our typos, and we will fix them in the revised version of our paper.
>
>  We thank you again for your insightful comments.

---

### Official Review · Reviewer_yyRt · 2025-03-12

**Overall Recommendation:** 2

**Summary:**

This paper studies binary hypothesis testing in the setting of softmax models and leverage score models. That is, quantifying the number of queries needed to identify an unknown distribution given two possible candidates. Some theoretical analysis shows the lower and upper bound for such problem.

**Claims And Evidence:**

1. In the introduction of the paper, authors claim that the paper is motivated by distinguish different ability parts of LLMs by limited parameters sampling. However, in the latter sections of the paper, the contents are not cycling back to the theme.

2. In addition, the claim in the first page "As we delve deeper .... self-attention" needs reference to support.

3. Authors study a softmax models and leverage score models which can be formulated as a matrix. However, it is not clear how the conclusions made on this single-layer model can be generalized to the LLM and transformer as a whole system.

4. The discussion on leverage scores appears forced and disconnected from the original motivation, aside from some vague remarks about their usefulness. While highlighting similarities may be valuable, the authors need to clarify how this perspective relates to the central question at hand.

**Essential References Not Discussed:**

No

**Experimental Designs Or Analyses:**

No experiment included in this paper.

**Methods And Evaluation Criteria:**

This paper is built based on theoretical analysis without any experiments.

**Other Comments Or Suggestions:**

No

**Other Strengths And Weaknesses:**

The paper presents no experiment to suggest the application scenario of the study, which significantly limit its influence for LLM/transformer community.

**Questions For Authors:**

See above

**Relation To Broader Scientific Literature:**

This paper relates more to binary hypothesis testing. Although, in the introduction, authors try to connect this paper with LLMs, no clear analysis has been made in the rest parts.

**Theoretical Claims:**

Yes.

---

> ### Author Rebuttal · Authors · 2025-04-01
>
> We express our deepest gratitude to the reviewer for the time and effort in reviewing our work. Below, we want to respond to the weaknesses and questions.
>
> **Concern 1**: However, in the latter sections of the paper, the contents are not cycling back to the theme.
>
> **Answer**: We thank you very much for your insightful comments. In Sections 3 and 4, we study the distinguishability of models (softmax and leverage score based) through the lens of binary hypothesis testing, establishing tight sample complexity bounds. These results directly address the challenge of determining how much information (or how many queries) is needed to tell apart closely related models, a theoretical formulation aligned with understanding model "abilities" via limited parameter access.
> Moreover, our framework sets a path toward identifying distinguishable components of large models. For instance, showing that certain parameters contribute more significantly to distinguishability (via Hellinger distance bounds) offers insight into what might constitute an “ability region” within a model — aligning with the introductory motivation.
>
> **Concern 2**: In addition, the claim in the first page "As we delve deeper .... self-attention" needs reference to support.
>
> **Answer**: We completely agree with your comment. In the revised version of our paper, we will include the following citations:
>
> $\bullet$ [1] introduced the Transformer model and the use of softmax in computing attention weights.
>
> $\bullet$ [2] discussed variants and performance implications of softmax in attention computation.
>
> $\bullet$ [3] and [4] directly analyze optimization on the softmax regression problem and show the impact of the softmax unit on self-attention and in-context learning.
>
>
> **Concern 3**: Authors study a softmax models and leverage score models which can be formulated as a matrix. However, it is not clear how the conclusions made on this single-layer model can be generalized to the LLM and transformer as a whole system.
>
> **Answer**: We thank you for your insightful comments and kindly refer you to our **Response to Weakness 1** to Reviewer 8u1R.
>
>
> **Concern 4**: The discussion on leverage scores appears forced and disconnected from the original motivation, aside from some vague remarks about their usefulness. While highlighting similarities may be valuable, the authors need to clarify how this perspective relates to the central question at hand.
>
>
> **Answer**: ​​Our primary motivation is to understand how model components, especially those arising in LLMs, can be distinguished via limited parameter sampling. While the softmax mechanism arises directly in self-attention, the leverage score model serves as a broader, more general abstraction of distributional behavior driven by matrix-parameterized functions, similar in form to softmax. Both softmax and leverage score models define distributions over outputs conditioned on structured inputs, and both are parameterized by matrices, making them amenable to a unified theoretical treatment through hypothesis testing.
> Rather than being a disconnected addition, the leverage score model allows us to extend our analysis framework and highlight that the difficulty of distinguishing close models under sample constraints is not unique to softmax, but also arises in other distributional settings relevant to algorithms and data analysis.
>
> We thank you again for your insightful comments.
>
>
>
> [1] Ashish Vaswani, Noam Shazeer, Niki Parmar, Jakob Uszkoreit, Llion Jones, Aidan N. Gomez, Lukasz Kaiser, and Illia Polosukhin. "Attention is all you need." NeurIPS’17.
>
> [2] Krzysztof Choromanski, Valerii Likhosherstov, David Dohan, Xingyou Song, Andreea Gane, Tamas Sarlos, Peter Hawkins et al. "Rethinking attention with performers." ICLR’21.
>
> [3] Zhihang Li, Zhizhou Sha, Zhao Song, Mingda Wan. "Attention scheme inspired softmax regression." ICLR’25 workshop.
>
> [4] Yeqi Gao, Zhao Song, and Junze Yin. "An iterative algorithm for rescaled hyperbolic functions regression." AISTATS’25.

---

### Official Review · Reviewer_8u1R · 2025-03-14

**Overall Recommendation:** 3

**Summary:**

The paper derived orderwise tight upper and lower bounds on the sample complexity of hypothesis testing for softmax distributions (capturing the last layer output) and the leverage score distribution.

**Claims And Evidence:**

Yes

**Essential References Not Discussed:**

N/A

**Experimental Designs Or Analyses:**

N/A

**Methods And Evaluation Criteria:**

Yes

**Other Comments Or Suggestions:**

Typo in "retrieval argument generation (RAG)" (should be "augmented")

**Other Strengths And Weaknesses:**

Strengths: The paper derived tight upper and lower bounds on sample complexity of hypothesis for softmax and leverage score distributions, respectively, which are interesting and solid results.

Weaknesses: While both softmax and leverage score functions have many applications, it would be good to motivate more about the hypothesis testing on these two types of distributions. Also in neural networks, there can be multiple layers. How the analysis in the paper extends to multiple layers of softmax functions or how the single layer softmax fits in such cases.

Similarly for leverage score distributions. I am not familiar with the leverage score applications. Are there motivations for hypothesis testing for leverage score distributions?

In terms of proof techniques, is it possible to elaborate more on the challenge or novelty part of the proof?

**Questions For Authors:**

See above.

**Relation To Broader Scientific Literature:**

Softmax distributions are highly common in neural networks and the hypothesis testing of such distributions might be of interest.

**Theoretical Claims:**

Looks correct.

---

> ### Author Rebuttal · Authors · 2025-04-01
>
> We express our deepest gratitude to the reviewer for the time and effort in reviewing our work. Below, we want to respond to the weaknesses and questions.
>
> **Response to Weakness 1**: We thank the reviewer for raising this important point. Our work intentionally focuses on single-layer softmax and leverage score models as a foundational step in the theory of binary hypothesis testing over these widely-used classes of distributions. The novelty of our contribution lies in formalizing and analyzing the binary hypothesis testing problem in these structured settings, which, to the best of our knowledge, has not been previously addressed in the literature.
>
> Regarding motivation, we highlight that both softmax and leverage score distributions arise naturally in machine learning and numerical linear algebra, but their sample complexity under hypothesis testing had remained unexplored. We clarify this explicitly in the introduction (Section 1) and further elaborate on the motivation from LLMs and self-attention mechanisms, where softmax distributions are central.
>
> As for the extension to multiple layers of softmax, we agree that this is an important direction and appreciate the opportunity to clarify. In deep neural networks, softmax layers often appear at the output or in attention blocks. While our current theoretical framework addresses a single softmax layer parameterized by matrix $A$, multi-layer architectures can be viewed as compositions of functions, where only the final layer produces a distribution over outputs (which can be seen in Figure 1 of the famous “attention is all you need” paper https://arxiv.org/pdf/1706.03762). Therefore, the final softmax layer can still be abstracted and analyzed using our current hypothesis testing framework.
>
> In the revised version of our paper, we plan to discuss this direction more explicitly and include formal remarks about the potential for generalization to multi-layer or multi-head softmax settings.
>
> **Response to Weakness 2**: Thank you very much for your insightful comments. Leverage scores are a central concept in numerical linear algebra, machine learning, and graph algorithms. As detailed in Section 1 and Section 4 of our paper, they appear in numerous algorithmic applications including matrix approximation (e.g., CUR decomposition), randomized linear algebra, graph sparsification, maximum flow and matching, and random spanning tree generation.
>
> In all these applications, a leverage score distribution defines a probability distribution over data points, rows, or graph elements, used for randomized sampling or importance weighting. Given their probabilistic nature and sensitivity to the underlying data matrix, it is natural to ask whether two leverage score models correspond to the same or different underlying structures, particularly when the models are accessed as black boxes, which motivates our formulation of the binary hypothesis testing problem for leverage score distributions.
>
> **Response to Weakness 3**: The core novelty of our proofs lies in adapting classical binary hypothesis testing, typically studied in the context of generic distributions, to the structured, parameterized families of distributions induced by softmax and leverage score models. Unlike arbitrary distributions, these models produce distributions that are nonlinear functions of the input and matrix parameters, which poses unique analytical challenges.
>
> For the softmax model, one key difficulty is that different parameter matrices $A$ and $B$ can induce indistinguishable distributions due to invariance under certain transformations (e.g., row shifts). To handle this, we introduce structural constraints (e.g.,$\\|A - B\\|\_{2 \to \infty}$) and prove that the Hellinger distance between softmax outputs under constrained inputs governs the sample complexity. This leads to a tight upper and lower bound framework via careful analysis of the sensitivity of softmax distributions to perturbations in the parameter matrix.
>
> For the leverage score model, the challenge is even greater due to the nonlinear matrix expressions involved, including matrix inversion and normalization, and the fact that the input sss is a vector that rescales rows of the matrix. We overcome this by establishing operator norm bounds and using perturbation theory to relate changes in the parameter matrix to changes in the output distribution. Our proof carefully propagates these changes through the matrix expressions and yields tight dependence on both the model difference $\\|A - B\\|$ and input constraints.
>
> **Response to the question**: Thank you very much for carefully checking this. We will fix this in the revised version of our paper.
>
> We thank you again for your insightful comments.

---

### Decision · Program_Chairs · 2025-05-01

**Decision:**

Accept (poster)

**Comment:**

This paper provides sample complexity bounds for hypothesis testing of softmax and leverage score distributions. The theoretical results look correct. While authors argue that these results are a step towards understanding power of different parts of language models, the reviewers were not convinced. Ultimately, the latter is a subjective belief but for topics that are not central to the conference, it is worthwhile to take into account the feedback of reviewers regarding relevance. Therefore, my recommendation is weak acceptance.